# HOW TO DECAY YOUR LEARNING RATE

## ABSTRACT

Complex learning rate schedules have become an integral part of deep learning. We find empirically that common fine-tuned schedules decay the learning rate after the weight norm bounces. This leads to the proposal of ABEL: an automatic scheduler which decays the learning rate by keeping track of the weight norm. ABEL's performance matches that of tuned schedules, is more robust with respect to its parameters and does not depend on the time budget. Through extensive experiments in vision, NLP, and RL, we show that if the weight norm does not bounce, we can simplify schedules even further with no loss in performance. In such cases, a complex schedule has similar performance to a constant learning rate with a decay at the end of training.

## 1 INTRODUCTION

Learning rate schedules play a crucial role in modern deep learning. They were originally proposed with the goal of reducing noise to ensure the convergence of SGD in convex optimization (Bottou, 1998). A variety of tuned schedules are often used, some of the most common being step-wise, linear or cosine decay. Each schedule has its own advantages and disadvantages and they all require hyperparameter tuning. Given this heterogeneity, it would be desirable to have a coherent picture of when schedules are useful and to come up with good schedules with minimal tuning.

While we do not expect dependence on the initial learning rate in convex optimization, large learning rates behave quite different from small learning rates in deep learning (Li et al., 2020a; Lewkowycz et al., 2020). We expect the situation to be similar for learning rate schedules: the non-convex landscape makes it desirable to reduce the learning rate as we evolve our models. The goal of this paper is to study empirically (a) in which situations schedules are beneficial and (b) when during training one should decay the learning rate. Because of the use of stochastic gradients, the loss will have a positive definite contribution due to noise. While this noise is key to explore the landscape, we want to minimize this contribution at the end of training and we will use a *simple* schedule as our baseline: a constant learning rate with one decay close to the end of training. Training with this smaller learning rate for a short time is expected to reduce the noise without letting the model explore the landscape too much. This is corroborated by the fact that the minimum test error often occurs almost immediately after decaying the learning rate. Part of the paper focuses on comparing the *simple* schedule with standard *complex* schedules used in the literature, studying the situations in which these complex schedules are advantageous. We find that complex schedules are considerably helpful whenever the weight norm bounces, which happens often in the usual, optimal setups. We observe that the weight norm (sum over the squared $L_2$-norm of the weight in each layer) often presents a bouncing behaviour: it decreases, hits a minimum after and continues increasing. In the presence of a bouncing weight norm, we propose an automatic scheduler which performs as well as fine-tuned schedules.

### 1.1 OUR CONTRIBUTION

The goal of the paper is to study the benefits of learning rate schedules and when the learning rate should be decayed. We focus on the dynamics of the weight norm (sum over layers of the square of weights' $L_2$-norms).

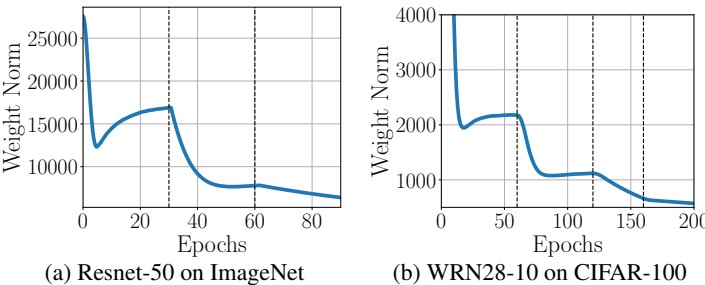

(a) Resnet-50 on ImageNet    (b) WRN28-10 on CIFAR-100

Figure 1: Evolution of the weight norm when training with step-wise decay (decay times marked by black dashed lines). The learning rate is decayed when the weight norm converges after bouncing. Models were evolved in optimal settings whose tuning did not use the weight norm as input.

The paper is divided into two parts and their main conclusions are:

1. **Decaying the learning rate using complex schedules is only beneficial in the presence of weight bouncing.** $L_2$ regularization seems crucial for the weight norm to bounce and in its absence (which is common in NLP and RL) we don't see a benefit from complex schedules. This is explored in detail in section 3 and the results are summarized in table 2.

2. **In the presence of bouncing, one should use the weight norm to inform when to decay the learning rate.** We observe that tuned step-wise schedules decay the learning rate when the weight norm converges after bouncing[1]. Towards the end of training, a last decay decreases the noise. See figure 1. We propose an *Automatic, Bouncing into Equilibration Learning rate scheduler* (ABEL). ABEL is competitive with fine-tuned schedules and needs less tuning (see table 1 and discussion in section 2).

While these two points are somewhat independent, together they provide a very general picture of when to decay the learning rate. For pedagogical purposes, we present the two points in inverse order: first reverse engineer what makes fine-tuned learning rate schedules work and then study to what extent these are helpful across tasks and domains.

**The origin of weight bouncing.**    In section 4, we explain why the weight norm often bounces.

**Weight bouncing and performance.**    Generally speaking, weight bouncing occurs when we have non-zero $L_2$ regularization and large enough learning rates. While $L_2$ regularization is crucial in vision tasks, it is not found to be that beneficial in NLP or Reinforcement Learning tasks (for example Vaswani et al. (2017) does not use $L_2$). If the weight norm does not bounce, ABEL yields the "simple" learning rate schedule that we expect naively from the noise reduction picture: decay the learning rate once towards the end of training. We confirm that, in the absence of bouncing, such a simple schedule is competitive with more complicated ones across a variety of tasks and architectures, see table 2. We also see that the well known advantage of momentum compared with Adam for image classification in ImageNet (see Agarwal et al. (2020) for example) seems to disappear in the absence of bouncing, when we turn off $L_2$ regularization. Weight norm bouncing thus seems empirically a necessary condition for non-trivial schedules to provide a benefit, but it is not sufficient: we observe that when the datasets are easy enough that simple schedules can get zero training error, schedules do not make a difference for the generalization performance (models can hit zero training error and still improve the test error see Lewkowycz & Gur-Ari (2020) for example).

## 1.2    RELATED WORKS

We do not know of any explicit discussion of weight bouncing in the literature. The dynamics of deep networks with $L_2$ regularization has drawn recent attention, see for example van Laarhoven

---

[1]We define bouncing by the monotonic decrease of the weight norm followed by a monotonic increase which occurs for a fixed learning rate as can be seen in figure 1.

(2017); Lewkowycz & Gur-Ari (2020); Li et al. (2020b); Wan et al. (2020); Kunin et al. (2020). The recent paper Wan et al. (2020) observes that the weight norm equilibration is a dynamical process (the weight norm still changes even if the equilibrium conditions are approximately satisfied) which happens soon after the bounce.

The classic justification for schedules comes from reducing the noise in a quadratic potential (Bottou, 1998). Different schedules do not provide an advantage in convex optimization unless there is a substantial mismatch between train and test landscape (Nakkiran, 2020), however this is not the effect that we are observing in our setup: when schedules are beneficial, their training performance is substantially different, see for example figure S4. Of course, this is not too surprising because convex optimization does not apply to deep networks. The work of Li et al. (2020a) could be helpful to understand better the theory behind our phenomena, although it is not clear to us how their mechanism can generalize to multiple decays (or other complex schedules). There has been lots of empirical work trying to learn schedules/optimizers see for example Maclaurin et al. (2015); Li & Malik (2016); Li et al. (2017); Wichrowska et al. (2017); Rolinek & Martius (2018); Qi et al. (2020); You et al. (2017, 2020). Our approach does not have an outer loop: the learning rate is decayed depending on the weight norm, which is conceptually similar to the `ReduceLROnPlateau` scheduler, where the learning rate is decayed when the loss plateaus which is present in most deep learning libraries. However, `ReduceLROnPlateau` does not perform well across our tasks. A couple of papers which thoroughly compare the performance of learning rate schedules are Shallue et al. (2019); Kaplan et al. (2020).

## 2 AN AUTOMATIC LEARNING RATE SCHEDULE BASED ON THE WEIGHT NORM

### ABEL AND ITS MOTIVATION

From the two setups in figure 1 it seems that optimal schedules tend to decay the learning rate after bouncing, when the weight norm growth slows down[2]. We can use this observation to propose ABEL (Automatic Bouncing into Equilibration Learning rate scheduler): a schedule which implements this behaviour automatically, see algorithm 1. In words, we keep track of the changes in weight norm between subsequent epochs (denoted by $t$), $\Delta|w_t|^2 \equiv |w_t|^2 - |w_{t-1}|^2$. When the sign of $\Delta|w_t|^2$ flips, it necessarily means that it has gone through a local minimum: because $\Delta|w_t|^2 < 0$ initially if $\Delta|w_{t+1}|^2 \cdot \Delta|w_t|^2 < 0, \Delta|w_t|^2 < 0$, then $|w_t|$ is a minimum: $|w_{t+1}| > |w_t| < |w_{t-1}|$. After this, the weight norm grows and slows down until at some point $\Delta|w_t|^2$ is noise dominated. In this regime, $\Delta|w_t|^2$ will become negative, which we will take as our decaying condition. In order to reduce SGD noise, near the end of training we decay it one last time. In practice we do this last decay at around $85\%$ of the total training time and as we can see in the SM B.1, this particular value does not really matter as long as it is a few epochs. This schedule does not require a fixed number of epochs (as opposed to cosine decay which is strict in this regard): one can decide to train models for longer times by loading the checkpoint before this last noise reducing decay, or stop models early (and do the corresponding last decay for a few epochs).

Algorithm 1 is an implementation of the idea with the base learning rate and the decay factor as the main hyperparameters. While alternative implementations could be more explicit about the weight norm slowing down after reaching the minimum, they would likely require more hyperparameters. The definition 1 does not guarantee that the learning rate will be decayed after the weight norm reaches a minimum, in particular if the weight norm was noisy, 1 could trigger a learning rate decay without bouncing. In practice, in all the experiments that we ran (which span variations across 4 datasets, 6 learning rates, 5 decay factors, 4 architectures), ABEL successfully decayed the learning rate after the weight norm reached equilibrium.

We have decided to focus on the total weight norm, but one might ask what happens with the layer-wise weight norm. In SM B.2, we study the evolution of the weight norm in different layers. We focus on the 10 layers which contribute the most to the weight norm (these layers account for $50\%$ of the weight norm). We see that most layers exhibit the same bouncing plus slowing down pattern as the total weight norm and this happens at roughly the same time scale.

---

[2]The decay rate and base learning rates are not the same between experiments, but here we focus on when the learning rate decay occurs.

---

**Algorithm 1** ABEL Scheduler

---

# Repeat every epoch $t$, reached_minimum=False initially.
**if** $(|w_t|^2 - |w_{t-1}|^2) \cdot (|w_{t-1}|^2 - |w_{t-2}|^2) < 0$ **then**
    **if** reached_minimum **then**
        learning_rate = decay_factor $\cdot$ learning_rate
        reached_minimum=False
    **else**
        reached_minimum=True
    **end if**
**end if**
# If no fixed budget, decay for a few epochs when training is done instead, see end of sec.2.
**if** t = last_decay_epoch **then**
    learning_rate = decay_factor $\cdot$ learning_rate
**end if**

---

| Setup | | Test error | | |
|---|---|---|---|---|
| Dataset | Architecture | Step-wise | ABEL | Cosine |
| ImageNet | Resnet-50 | 24.0 | 23.8 | 23.2 |
| CIFAR-10 | WRN 28-10 | 3.7 | 3.8 | 3.5 |
| CIFAR-10 | VGG-16 | - | 7.1 | 6.9 |
| CIFAR-100 | WRN 28-10 | 18.5 | 18.7 | 18.4 |
| CIFAR-100 | PyramidNet | - | 10.8 | 10.8 |
| SVHN | WRN 16-8 | 1.77 | 1.79 | 1.89 |

Table 1: Comparison of test error at the end of training for different setups and learning rate schedules. We see that ABEL has very similar performance to the fine-tuned step-wise schedule without the need to tune when to decay. ABEL uses the baseline values of learning rates and decay factors and we have not fine-tuned these. The cells denoted by - refer to setups for which we do not have reference step-wise decays. Step-wise and cosine schedules use the optimal hyperparameters from our baselines and we plug these values for ABEL without further tuning, see SM for more experimental details.

PERFORMANCE COMPARISON ACROSS SETUPS

We have run a variety of experiments comparing learning rate schedules with ABEL, see table 1 for a summary and figure 2 for some selected training curves ( rest of the training curves are in SM D.1). We use ABEL without hyperparameter tuning: we are plugging the base learning rate and the decay factor of the reference step-wise schedule (these reference decay factors are $0.2$ for CIFAR and $0.1$ for other datasets). We see that ABEL is competitive with existing fine-tuned learning rate schedules and slightly outperforms step-wise decay on ImageNet. Cosine often beats step-wise schedules, however as we will discuss shortly, such decay has several drawbacks.

ROBUSTNESS OF ABEL

ABEL is quite robust with respect to the learning rate and the decay factor. Since it depends implicitly on the natural time scales of the system, it will adapt to when to decay the learning rate. We can illustrate this by repeating the ImageNet experiment with different base learning rates or decay factors. The results are shown in figure 3. Note: when the decay factor is $0.5$ we evolved both models for 120 epochs (denote with star in figure) to allow more time to bounce. We observe more bounces for larger decay rates and shallower bounces the smaller the learning rate.

We would like to highlight the mild dependence of performance on the learning rate: if the learning rate is too high, the weight norm will bounce faster and ABEL will adapt to this by quickly decaying the learning rate. This can be seen quite clearly in the learning rate $= 16$ training curves, see SM D.4.

ABEL also has the 'last_decay_epoch' hyperparameter, which determines when to perform the last decay in order to reduce noise. Performance depends very weakly on this hyperparameter (see SM more) and for all setups in table 1 we have chosen it to be at $85\%$ of the total training time. The most

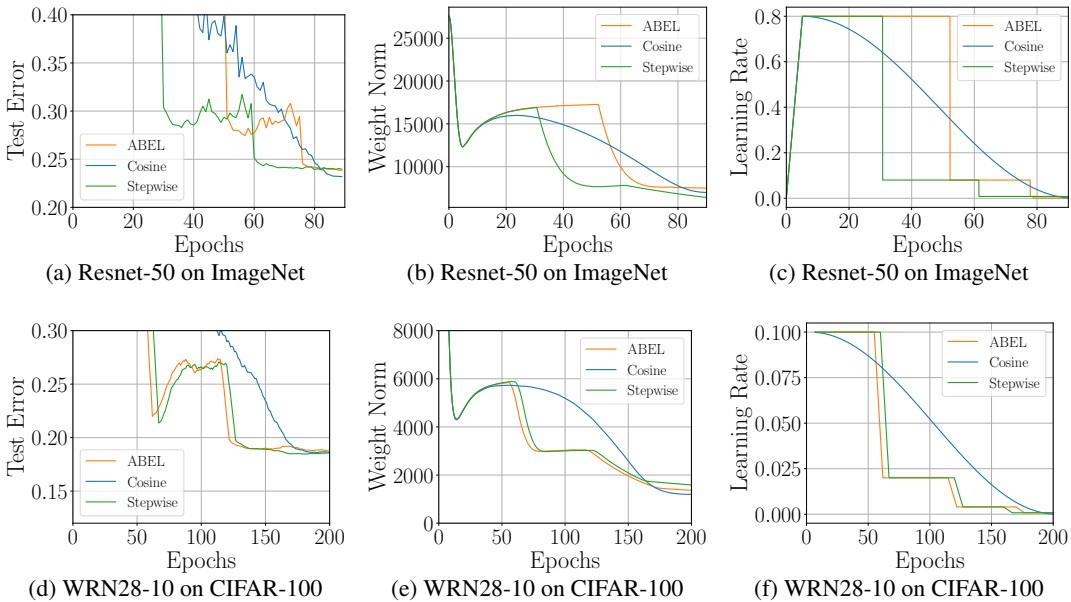

Figure 2: Training curves of two experiments from table 1.

natural way to think about this would be to run ABEL for a fixed amount of time and after decay the learning rate for a small number of epochs in order to get the performance with less SGD noise.

COMPARISON OF ABEL WITH OTHER SCHEDULES

It is very natural to compare ABEL with step-wise decay. Step-wise decay is complicated to use in new settings because on top of the base learning rate and the decay factor, one has to determine when to decay the learning rate. ABEL, takes care of the 'when' automatically without hurting performance. Because when to decay depends strongly on the system and its current hyperparameters, ABEL is much more robust to the choices of base learning rate and decay factor.

A second class of schedules are those which depend explicitly on the number of training epochs ($T$), like cosine or linear decay. This strongly determines the decay profile: with cosine decay, the learning rate will not decay by a factor of 10 with respect to its initial value until 93% of training! Having $T$ as a determining hyperparameter is problematic: it takes a long time for these schedules to have comparable error rates to step-wise decays, as can be seen in figures 2, S8. This implies that until very late in training one can not tell whether $T$ is too short, in which case there is no straightforward way to resume training (if we want to evolve the model with the same decay for a longer time, we have to start training from the beginning). This is part of the reason why large models in NLP and vision use schedules which can be easily resumed like rsqrt decay (Vaswani et al., 2017), "clipped" cosine decay (Kaplan et al., 2020; Brown et al., 2020) or exponential decay (Tan & Le, 2020). In contrast, for ABEL the learning rate at any given time is independent of the total training budget ( while there is the last_decay_epoch parameter, it can easily be evolved for longer if we load the model before the last decay).

We have decided to compare ABEL with the previous two schedules because they are the most commonly used ones. There are a lot of automatic/learnt learning rate schedules (or optimizers), see Maclaurin et al. (2015); Li & Malik (2016); Li et al. (2017); Wichrowska et al. (2017); Yaida (2018); Rolinek & Martius (2018); Qi et al. (2020) and to our knowledge most of them require either significant change in the code (like the addition of non-trivial measurements) or outer loops and also add hyperparameter of their own, so these are never completely hyperparameter free. Compared with these algorithms ABEL is simple, interpretable (it can be easily compared with fine-tuned step-wise decays) and performs as well as tuned schedules. It is also quite robust compared with other automatic

methods because it relies in the weight norm which is mostly noise free through training compared with other batched quantities like gradients or losses.

An algorithm similar in simplicity and interpretability is `ReduceLROnPlateau` which is one of the basic optimizers of PyTorch or TensorFlow and decays the learning rate whenever the loss equilibrates. We train a Resnet-50 ImageNet model and a WRN 28-10 CIFAR-10 model with this algorithm , see SM B.3 for details. We use the default hyperparameters and for the ImageNet experiment, the learning rate does not decay at all, yielding a test error of 47.4. For CIFAR-10, `ReduceLROnPlateau` does fairly well, test error of 3.9, however the learning rate decays without bound rather fast. These two experiments suggest that `ReduceLROnPlateau` can not really compete with the schedules described above: it seems rather finicky with respect to hyperparameters and we do not want to have to precisely tune different hyperparameters for a given setup (see robustness discussion).

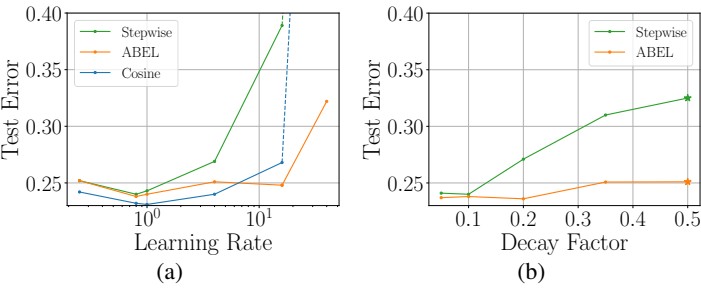

Figure 3: ResNet-50 trained on ImageNet for different learning rates and decay factors. (a) ABEL beats others schedules when using non-optimal learning rates. At learning rate 40, only ABEL converges. (b) ABEL is robust with respect to changes in the decay factor, its performance does not depend too much on the decay factor because it adjusts the number of decays accordingly.

ABEL DOES NOT REQUIRE A FIXED TRAIN BUDGET

From the empirical studies, the drop in the test error after decaying the learning rate is upper bounded by the previous drops in the test error, a reason for this is that this drop can be attributed to a reduction of the SGD noise and smaller learning rates have less SGD noise. This provides an automatic way of prescribing the train budget: if the improvement of accuracy after a decay is smaller than some threshold, exit training after a small number of epochs (to process the last decay). This approach does not have 'last_decay_epoch' hyperparameter. Such a training setup would not be possible for cosine/linear decay by construction since they depend on the training budget. This seems hard for step-wise decay since there is no way to predetermine how to decay the learning rate automatically. In order to compare ABEL with the standard baselines, all the experiments in the paper have been trained with a fixed budget.

## 3 SCHEDULES AND PERFORMANCE IN THE ABSENCE OF A BOUNCING WEIGHT NORM

In this section, we study settings where the weight norm does not bounce to understand the impact of learning rate schedules on performance. Setups without $L_2$ regularization are the most common situation with no bouncing, these setups often present a monotonically increasing weight norm. It is not clear to us what characteristics of a task make $L_2$ regularization beneficial but as shown in Table 2 it seems that Vision benefits considerably more from it than NLP or RL.

We conduct an extensive set of experiments in the realms of vision, NLP and RL and the results are summarized in table 2. In these experiments, we compare complex learning rate schedules with a simple schedule where the model is evolved with a constant learning rate and decayed once towards the end of training. This simple schedule mainly reduces noise: the error decreases considerably immediately after decaying and it does not change much afterwards (it often increases). Across these experiments, we observe that complicated learning rate schedules are not significantly better than the simple ones. For a couple of tasks (like ALBERT fine-tuning or WRN on CIFAR-100),

complex schedules are slightly better ($\sim 0.3\%$) than the simple decay but this small advantage is nothing compared with the substantial advantage that schedules have in vision tasks with $L_2$. Another situation where there is no bouncing weight norm is for small learning rates, for example VGG-16 with learning rate $0.01$, in such case there is also no benefit from using complex schedules, see SM B.5 for more details. Note that in this paper we are using $L_2$ regularization and weight decay interchangeably: what matters is that there is explicit weight regularization. These experiments also show that the well known advantage of momentum versus Adam for vision tasks is only significant in the presence of $L_2$. In the absence of a benefit from $L_2$ regularization / weight decay it seems like Adam is a better optimizer, Agarwal et al. (2020) suggested that this is because it can adjusts the learning rate of each layer appropriately and it would be interesting to understand whether there is any connection between that and bouncing.

These experiments have a growing weight norm as can be seen in SM D.3. While the weight norm does not have to be always increasing in the absence of $L_2$ regularization, this is a function of the learning rate (see section 4.1) , and the learning rates used in practice exhibit this property. Homogeneous networks with cross entropy loss will have an increasing weight norm at late times, see Lyu & Li (2020). Even if a simple schedule is competitive this does not imply that other features of convex optimization like the independence of performance in the learning rate carry over. We repeat the CIFAR-100 experiments for a fixed small learning rate of $0.02$ (the same as the final learning rate for the simple schedule) and the error with $L_2 = 0$ is $23.8$ while with $L_2 \neq 0$ is $29.7$, we see that while there is a performance gap between a small and large learning rates, this gap is much smaller if there is no bouncing (difference in error rate of $1.2\%$ for $L_2 = 0$ vs $7.5\%$ for $L_2 \neq 0$). For a fair comparison with small learning rates, we evolved these experiments for $5$ times longer than the large learning rates, but this did not give any benefit.

While the experiments presented in table 2 do not have $L_2$ regularization, some NLP architectures like Devlin et al. (2019); Brown et al. (2020) have weight decays of $0.01, 0.1$ respectively. We tried adding weight decay to our translation models and while performance did not change substantially, we were not able to get a bouncing weight norm.

The effect of different learning rate schedules in NLP was also studied thoroughly in appendix D.6 of Kaplan et al. (2020) with the similar conclusion that as long as the learning rate is not small and is decayed near the end of training, performance stays roughly the same.

**The presence of a bouncing weight norm does not guarantee that schedules are beneficial.** From this section, a bouncing weight norm seems to be a necessary condition for learning rate schedules to matter, but it is not a sufficient condition. Learning rate schedules seem only advantageous if the training task is hard enough. In our experience, if the training data can be memorized with a simple learning rate schedule before the weight norm has bounced, then more complex schedules are not useful. This can be seen by removing data augmentation in our Wide Resnet CIFAR experiments, see figure 4. In the presence of data augmentation, simple schedules can not reach training error $0$ even when evolved for 200 epochs, see SM.

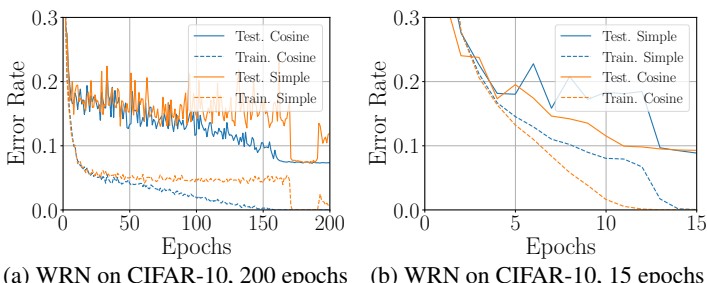

(a) WRN on CIFAR-10, 200 epochs    (b) WRN on CIFAR-10, 15 epochs

Figure 4: Wide Resnet on CIFAR-10 without data augmentation evolved for 200 epochs (left) and 15 epochs (right). In this setup, the weight norm bounces at around 15 epochs. a) Both schedules reach training error 0 and their performance is the same (error of 7.3). b) If we evolve the model for only 15 epochs, both schedules can still get training error 0 without a weight norm bounce, we think this is the reason why there is no performance difference in a).

| Setup | | | Performance for different schedules | |
|---|---|---|---|---|
| Type | Task and metric | Architecture | Complex Decay | Simple Decay |
| NLP | EN-DE, BLEU | Transformer | 29.0 | 28.9 |
| | EN-FR, BLEU | Transformer | 43.0 | 43.0 |
| | GLUE, Average score | ALBERT fine-tuning | 83.1 | 82.9 |
| RL | Qbert, Score | PPO | 1750 | 1850 |
| | Seaquest, Score | PPO | 21.0 | 20.7 |
| | Pong, Score | PPO | 22300 | 23000 |
| Vision $L_2 = 0$ | ImageNet, Test Accuracy | Resnet-50 | 71.9 | 72.2 |
| | ImageNet, Test Accuracy | Resnet-50 + Adam | 71.8 | 71.1 |
| | CIFAR-10, Test Accuracy | Wide Resnet 28-10 | 95.0 | 95.0 |
| | CIFAR-100, Test Accuracy | Wide Resnet 28-10 | 78.2 | 77.9 |
| Vision $L_2 \neq 0$ (has bounce) | ImageNet, Test Accuracy | Resnet-50 | 76.8 | 71.5 |
| | ImageNet, Test Accuracy | Resnet-50 + Adam | 75.3 | 74.0 |
| | CIFAR-10, Test Accuracy | Wide Resnet 28-10 | 96.5 | 95.1 |
| | CIFAR-100, Test Accuracy | Wide Resnet 28-10 | 82.2 | 77.8 |

Table 2: Comparison of performance between a simple learning rate decay and a "complex" decay among tasks: "complex" means cosine decay for vision tasks and linear decay for NLP and RL. For NLP and RL tasks higher metrics imply better performance, while for vision tasks, lower error denotes better performance. None of these tasks (except for the vision task with $L_2$ used as a reference) have weight norm bouncing nor an advantage from non-simple schedules. We have averaged the RL tasks over 3 runs and their difference is compatible with noise. See S1 for the individual GLUE scores, as it is common we have omitted the problematic WNLI. We use test accuracy $= 100-$ test error, so that for all metrics in these table, higher is better.

## 4 UNDERSTANDING WEIGHT NORM BOUNCING

In this section, we will pursue some first steps towards understanding the mechanism behind the phenomena that we found empirically in the previous sections.

### 4.1 INTUITION BEHIND BOUNCING BEHAVIOUR

We can build intuition about the dynamics of the weight norm by studying its dynamics under SGD updates. Given the loss $L_t$ in the absence of $L_2$ regularization, let $g_t \equiv \frac{dL_t}{dw_t}$ be its gradient and $\eta, \lambda$ are the learning rate and $L_2$ regularization coefficient, one can easily derive:

$$
\begin{aligned}
\Delta w_{t+1} &= w_{t+1} - w_t = -\eta g_t - \eta\lambda w_t \\
\Delta |w_{t+1}|^2 &= \eta^2|g_t|^2 - (2 - \eta\lambda)\eta\lambda|w_t|^2 - 2(1 - \eta\lambda)g_t \cdot w_t \\
&\approx \eta^2|g_t|^2 - 2\eta\lambda|w_t|^2 - 2\eta g_t \cdot w_t + O(\eta^2\lambda)
\end{aligned}
\tag{1}
$$

where we have used that empirically $\eta\lambda \ll 1$. This equation holds layer by layer and we include batch-norm weights if present. In the absence of $L_2$ regularization, for large enough learning rates $(\eta > \frac{|g_t|^2}{2g_t \cdot w_t})$, this suggests that the weight norm will be increasing.

Equation 1 can be further simplified for scale invariant networks, which satisfy $g_t \cdot w_t = 0$, see for example van Laarhoven (2017) [3]. In the absence of such term, we see that the updates of the weight norm are determined by the relative values of the gradient and weight norm. If $\lambda = 0$ or $\eta$ is very small, the weight norm updates will have a fixed sign and thus there will not be bouncing. More generally, we expect that in the initial stages of training, the weight norm is large and its dynamics are dominated by the decay term. As it shrinks, the relative value of the gradient norm term becomes larger and it seems natural that at some point, it will dominate, making the weight norm bounce. This is also studied in Wan et al. (2020), where it is shown that after the bounce, the two terms in equation 1 are the same order and the weight norm "dynamically equilibrates" (although it can not stay constant

---
[3] Scale invariant networks are defined by network functions which are independent of the weight norm: $f(\alpha w) = f(w)$. The weight norm of these functions still affects its dynamics (Li & Arora, 2019).

because the gradient norm changes with time). While we expect the $g_t \cdot w_t$ to be non-zero in our setups, only layers which are not scale invariant would contribute to this term and roughly any layer before a BatchNorm layer is scale invariant so we expect this term to be smaller than the other two.

The only necessary condition for a model to have a bouncing weight norm seems to be that it has $L_2$ regularization and a large learning rate. Empirically, we have seen that different optimizers, losses and batch sizes can have a bouncing weight norm.

## 4.2 TOWARDS UNDERSTANDING THE BENEFITS OF BOUNCING AND SCHEDULES

To better understand this phenomenon, finding the simplest model that captures it would be desirable: we expect this to be generic as long as we have $L_2$ regularization and the learning rate is big enough. We believe that learning rate schedules being only advantageous for hard tasks (as we discussed in section 3) is the principal roadblock to find theoretically tractable models of these phenomena.

For bouncing setups, decaying the learning rate when the weight norm is equilibrating allows the weight decay term in equation 1 to dominate, causing the weight norm to bounce again. However, from equation 1, in the absence of $L_2$ decaying the learning rate can only slow down the weight norm equilibration process and this implies that the weights change more slowly, see SM. It seems like the combination of weight bouncing and decaying the learning rate might be beneficial because it allows the model to explore a larger portion of the landscape. Exploring this direction further might yield better insights to this phenomenon, perhaps building on the results of Wan et al. (2020); Kunin et al. (2020).

In the SM C, we do some extra experiments which explore properties of bouncing weight norm models.

**The disadvantage of decaying too early or too late.**   Waiting for the weight norm to bounce seems key to get good performance. Decaying too late might be harmful because the weight norm does not have enough time to bounce again, but it is not clear if that is bad. We run a simple VGG-5 experiment on CIFAR-100 and see that decaying too early significantly hurts performance and it is best to decay is after the weight norm has started slowing down its growth, before it is fully equilibrated.

**Dependence on initialization scale.**   One could wonder if the bounce would disappear if we change the initialization of the weights so that the initial weight norm is smaller than the minimum of the bounce with the original normalization. We studied this and conclude that even for very small initialization scales, there is a bouncing weight norm. If the initialization scale is too small, the bouncing weight norm disappears and the performance gets significantly degraded.

## 5 CONCLUSIONS AND LIMITATIONS

In this work we have studied the connections between learning rate schedules and the weight norm. We have made the empirical observation that a bouncing weight norm is a necessary condition for complex learning rate schedules to be beneficial, and we have observed that step-wise schedules tend to decay the learning rates when the weight norm equilibrates after bouncing. We have checked these observations across architectures and datasets and have proposed ABEL: a learning rate scheduler which automatically decays the learning rate depending on the weight norm, performs as well as fine-tuned schedules and is more robust than standard schedules with respect to its initial learning rate and the decay factor. In the absence of weight bouncing, complex schedules do not seem to matter.

We have studied a diverse yet fixed number of setups. For vision tasks where the weight norm bounces, ABEL provides an automatic schedule which has significant advantages in terms of hyperparameter tuning. The other main application is that without bouncing (common in NLP and RL), ABEL is a constant learning rate with a decay at the end of training and this matches the performance of complicated schedules. This might have been expected from the fact that different groups use different schedules in NLP (linear, rsqrt, clipped cosine) without significant impact in performance. In this context, we would like to argue that there is no need to optimize for the precise form of the schedules. Weight bouncing does not seem related with warmup and we have kept it if the baselines had it. ABEL does not improve the results of common schedules but it is quite robust and does not depend on the total number of epochs, making it more efficient when confronting new setups.

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
