# OpenReview forum: "How to decay your learning rate"
_ICLR.cc/2022/Conference — ICLR 2022 Submitted_

### Official Review · Reviewer_oLgq · 2021-10-20

**Correctness:** 4
**Technical Novelty And Significance:** 3
**Empirical Novelty And Significance:** 3
**Recommendation:** 6
**Confidence:** 4

**Main Review:**

Finding better learning rate schedulers, either in the metric of performance or robustness, is an important aspect of training deep networks. The suggestion by the authors to autotune the learning rate schedule by an internal property of the DNN (namely the bouncing feature of the weight norms) is interesting as well as the conjecture that such oscillations of the weight norm are a necessary condition for complex learning schedules. In addition, the equation governing this bouncing behaviour is simple enough to attract further theoretical interest. The scope of the experiments is also reasonable in my mind.

It would have been more exciting to show performances that exceed state of the art in some models, however, I accept that this is quite a high bar due to the mature state of the field. It remains to be seen whether this algorithm will be taken up by practitioners and whether it would push the state of the art in the field.

The work compares the ABEL scheduler with only a single adaptive learning rate scheduler (reduceLRonPlateau) and shows superiority. Other than that they compare with non-adaptive methods. What about algorithms such as AdaGrad or other adaptive schedulers on the market? More fundamentally, one can argue that their scheduling based on weight norm will be similar to scheduling based on any clear "equilibration" feature of the training loss. True the authors show that a plateau is not a robust feature. But still, I could imagine that, for instance, waiting for the average derivative of the training loss or the average gradient to decay to 10% of its initial value would be just as good. To summarize it'll be nice to disentangle whether it is the adaptive nature of the algorithm that is mainly useful here, or whether adapting it specifically based on the weight norms is central to its success.

More minor comments are -

1. In addition in their algorithm, I didn't see a precise treatment of a sticky point - what happens if one initializes the weights to a smaller value thereby reaching the maximum before the minimum? Put differently how to initialize the reached_minima variable, by measuring the slope of the weight_norm at the first epoch?

2. The paragraph which starts with "The origin of weight bouncing." tries both to explain what was the contribution and derive it in a hand-wavy manner. I'd personally be happier with just citing the contribution and deferring the arguments to where there's more room to go over them properly.

3. The authors comment that cosine annealing takes 93% of the training budget to reduce LR by a factor of 10 and finish the sentence with an exclamation mark. I don't see why that's necessarily an obvious drawback.

4. In the motivation section, the authors say "From the two setups in figure 1 it seems that optimal schedules tend to decay the learning rate after bouncing when the weight norm growth slows down." which can be taken to think that these two schedules really match. However, there's often a factor of 2 between them and occasionally (for Wide Resnet 16-8 on SVHN) a factor of 5.





**Summary Of The Paper:**

This work deals with the optimization of learning rate schedules for deep learning. It offers three main contributions

1. An equation relating the change in weight norm to the gradient norm, learning rate, weight decay, and the weight norm--- for weights that go into batch-norm. This equation provides intuition for the behaviour of this weight norm.

2. An algorithm and code for adjusting the learning rate by reducing it by a factor once the norm of the weights reach a maximum, then reducing it again by the same factor after about 85% of the total training time. The algorithm is benchmarked on about 14 learning tasks, within vision, RL, and NLP and shows similar results when compared to other complex learning rate schedulers algorithms. Being adaptive, their algorithm has at least one less hyperparameter compared to the competition and appears more robust.

3. Empirical tests of a conjecture that complex learning rates are only beneficial in the presence of extremums in the overall weight norms.



**Summary Of The Review:**

Finding better learning rate schedulers, either in the metric of performance or robustness, is an important aspect of training deep networks. The suggestion by the authors to autotune the learning rate schedule by an internal property of the DNN (namely the bouncing feature of the weight norms) is interesting as well as the conjecture that such oscillations of the weight norm are a necessary condition for complex learning schedules. In addition, the equation governing this bouncing behaviour is simple enough to attract further theoretical interest. The scope of the experiments is also reasonable in my mind, although further comparison with other adaptive algorithms could be useful.

---

> ### Author Response · Authors · 2021-11-09
> **Thanks for the review!**
>
> Thanks a lot for the nice review and the effort you put into it. We address your comments below and would really appreciate it if you considered raising the score if you found the rebuttal satisfactory.
>
>
> - Adaptive optimizers: we consider Adam optimizers in section 3 and they can't really be compared with momentum/SGD with learning rate schedules because they perform much worse for vision tasks.
>
> - Adaptation vs weight norm:  While we started exploring defining schedules from loss, gradients, etc we were not able to find anything that would work as well as the weight norm. Quantities like the training loss or gradient norm are usually monotonic, independently of weight norm bouncing, which seems crucial for schedules to make a difference. In summary, to our knowledge there is no other non-monotonic quantity that captures the bouncing behavior of the weight norm and we are totally open to the possibility that such quantity exists.
>
> - `What happens if one initializes the weights to a smaller value thereby reaching the maximum before the minimum?` We discuss this in section 4 and SM.C . To our surprise, decreasing the initial value of the weights pushes the minima to lower values (fig SM.7), one can't reach the minima from initialization (probably because the location of the minima also depends on the gradient norm).
>
> -  `I don't see why  [cosine decay takes 93% training budget to reach 0.1 LR] is necessarily an obvious drawback.` While this is clearly not a drawback in terms of final performance, this makes it really hard to tell how well a model trained with cosine decay is doing: for example in the first column of SM.8 cosine decay always seems to be losing up to the very end.
>
> - Have made the request changes around "The origin of weight bouncing."
>
> - Have clarified "From the two setups in figure 1 it seems that optimal schedules ..." with a footnote in the rebuttal.

---

### Official Review · Reviewer_y2w2 · 2021-11-01

**Correctness:** 2
**Technical Novelty And Significance:** 2
**Empirical Novelty And Significance:** 3
**Recommendation:** 3
**Confidence:** 4

**Main Review:**

Strengths
- Step-wise schedules have been proved effective in a number of contexts (especially in computer vision). The proposed scheduler is a very simple modification that allows to automatically determine the decay epoch without adding much complexity to the resulting pipeline.
- Linking the weight norm to the learning rate is a novel and interesting take on the topic of automatically tuning learning rate schedules...

Weaknesses
- ... however, at the moment the work seems almost entirely heuristic driven. The paper needs more work before being ready for publication, as I will detail below.
1. Following on what I mentioned in the summary, while I can understand that, empirically, changes of sign in the weight norm happen in concomitance with what the authors say, I do not think that it is proper to present the method as it is done at the moment. From what I can read and from the pseudocode of the algorithm, ABEL decays the learning rate after registering two changes in the direction in which the weight norm is progressing. This does not necessarily mean that the first change happens at a (local) minimum while the second is because of noise. Just to be fully clear, what I mean is that, in principle, the first change could also be a local maximum. Then, empirically you can observe what the authors say, but, in order to strengthen the position of the paper, I think it would be necessary to show some more organized results about this phenomenon. For instance, reporting something on this line: "over N experiments changing Y and Z conditions (e.g. varying the random seed, the initial learning rate, dataset, network architecture, etc), we observe that pair of changes in weight progression are in X% of the cases linked to [....]". In these experiments, particular care must be taken not to confuse the causal direction of the interplay between weight norm and learning rate.
2. About the point above, one of the phenomena that worries me the most is that changes could be linked to noise. In fact, I noticed that in the code there is also a parameter called `meas_freq` that counteracts the presence of noise in some settings. This is a hyperparameter of the method which seems not to be mentioned in the main paper, which is probably used for some of the experiments (e.g. looking at the last row of Figure S8). Could the authors explain this? How sensible is ABEL w.r.t. this hyperparamter?
3. I can agree that ABEL is not as linked to the training budget as the cosine scheduler is, but I substantially disagree with the claim that ABEL does not require (at all) a fixed training budget. The training budget is even a parameter of ABEL in the code. This is not a major concern in my eyes, but I think the claim must be played down from what it is currently written.
4. There is ample space to improve the clarity of the paper. Here below are some points about this:
  a) For starters, I do not think that the authors mention explicitly what $t$ denotes (I assume it to be epochs, but could be e.g. also iterations...). This is quite important, especially when paired with observation 2.
  b) The introductory discussion about the learning rate decay depicts a rather partial picture (e.g. decay comes up also in the convergence proofs of subgradent methods for non-smooth optimization).
  c) The the sentence  *Given that stochastic gradients are used in deep learning, we will use a simple schedule as our baseline: a constant learning rate with one decay close to the end of training.* is quite obscure to me. Why using stochastic gradients should imply that a simple schedule is a good baseline?
  d) The overall organization makes it difficult to pinpoint the actual contribution of the paper, as observations and experimental results are mixed with comments and parenthesis.
  e) Equation (1) lacks context. How is the loss defined? Are the authors considering networks that use batch norm or not (or both)? If yes, are the parameters of the batch norm layers included in w? Generally, the paper would greatly benefit from a clearer framing of the context. What is the typical setting that the authors have in mind and that ABEL is designed for?
5. The utility of (automatic) learning rate schedulers can be twofold: 1) improve on the state of the art, possibly starting from standard hyperparameter settings 2) ease the burden of hyperparameter tuning, especially when dealing with novel learning settings and datasets. As ABEL seems not to excel at 1) I think the authors should focus more on experiments to support 2). These could be done e.g. setting up time-controlled (repeated) experiments as in [1] .
6. I do not think that section 4 adds much value to the paper. Could the authors comment and summarize their findings reported in sec 4. and what do they add to e.g. Wan et al. (2020)?

Typos:
- which depend explicitly *in* (on) the number of training
- to have a bouncing weight norm is that *is* (it?) has L2 regularization

[1] Donini, Michele, et al. "MARTHE: Scheduling the Learning Rate Via Online Hypergradients." IJCAI 2020.

--------------------------------
**Post-rebuttal:** I thank the authors for their reply and modifications to the text that addressed some of my points. However, I remain of the opinion that this paper needs more work on several fronts before recommending acceptance. Among these, I would remark on: better treatment of the background material, clearer identification on when the weight norm behaviour happens beside $L^2$ norm (possibly looking also for counter-examples!), rethinking section 6, and a more convincing set of experiments (for showing convincing evidence about e.g. 5.2).
Regarding this last point, I want to clarify that in my review I mentioned [1] not for the grid search, but rather for the time-controlled experiments. If you go with random search for selecting the hyperparameters of the learning rate adaptation methods. I personally think that a recipe to make the comparison fair enough is to choose a prior distribution (e.g. uniform/log uniform) that covers reasonable values (e.g. as used for different datasets) with mean equal/close to the known well-performing ("optimal") value.


I think the main idea of the paper - i.e. linking weight norm to learning rate - is a quite interesting, novel and practical take. I do not raise my score because I think that the work would benefit greatly from a thorough review which could lead to a stronger submission to a later conference.

For 4.b, see e.g. Bertsekas, Dimitri P. (2015). Convex Optimization Algorithms (Second ed.). Belmont, MA.: Athena Scientific.

**Summary Of The Paper:**

This paper presents an empirical study of the dynamics of the total norm of the weights of neural network models in relation to the learning rate scheduling, focusing on what the authors call the "bouncing effect".
Based on empirical findings, they propose a method called ABEL to automatically decay the learning rate in a step-wise fashion. Essentially ABEL decays the learning rate after registering two changes in the direction in which the weight norm is progressing. The authors assume that the first change occurs at a local minimum, while the subsequent one indicates that the noise prevails the signal.
ABEL further decays the learning rate one last time at ~85% of the total training epochs.
They show that the cosine annealing and (manually) tuned step-wise schedules are beneficial when the bouncing effect happens, while in the absence of it, they claim that using a constant learning rate for most of the training epochs with a final decay is sufficient to obtain good performances.
The paper concludes with a section dedicated to understanding the bouncing effect which recalls some recent works that study the weight dynamics of neural nets.

**Summary Of The Review:**

Linking the weight norm to the learning rate is a novel and interesting take on the topic of automatically tuning learning rate schedules. However, at the moment the work seems almost entirely heuristic driven. The paper needs more work before being ready for publication, both in terms of writing and exposing ideas and in terms of experiments, especially regarding the main assumptions of ABEL.

---

> ### Author Response · Authors · 2021-11-09
> **Response to review**
>
> Thanks for the detailed and through review, we really appreciate the time you spent doing this. We address your comments below (and will fix the typos):
>
> 1/ What we want to do is "to decay the learning rate after equilibrium is reached after reaching a minima". We agree that the algorithm could fail to do that, specially if the weight norm curve was too noisy. In our experiments, ABEL did not have these issues, we added a sentence similar to what the reviewer proposes in the second paragraph of section 2. We picked this way of doing it to minimize hyperparameters, and there are probably other algorithms which can guarantee the learning rate is decayed when it is supposed to.
>
> 2/ We fixed `meas_freq` = 1epoch across experiments. This seemed to decay the learning rate at the right times when eye-balling the weight norm curve which is why we did not explore it further. Of course, if `meas_freq` were too small, noise would probably dominate the algorithm and the learning rate would not be decayed when it is supposed to.
>
> 3/ This is a fair point, the reason why we have used a training budget is because we want to compare with the standard baselines which use such setting, we added this clarification in page 6 before section 3. Would the reviewer agree that one could in principle train ABEL without a budget and stop training after some criteria for diminishing returns is reached?
>
> 4/ a) Yes, t means epochs, we mentioned this in page 3 but have made it more explicit in the algorithm and main text. b) We are not too familiar with those results and would be more than happy to include them in the manuscript if the reviewer can provide some references. c) We agree that it seems obscure and have tried to clarify this: it reads instead "Because of the use of stochastic gradients, the loss will
> have a positive definite contribution due to noise. While this noise is key to explore the landscape,
> we want to minimize this contribution at the end of training and we will use a simple schedule as
> our baseline: a constant learning rate with one decay close to the end of training". Our motivation comes from the quadratic linear model: there mini-batching forces the parameters to bounce around the local minima. One last decay substantially reduces the noisy part of the loss (in the quadratic setting we expect the noise contribution to the loss to scale like $\eta^2$ ). e) We have clarified the context around equation 1. We consider networks with and without batch norm and the batch norm parameters are also included in w, which we added to the text. We expect ABEL to be usable in any setting. Only if there is weight bouncing it will do something non-trivial (in the absence of bouncing it will just be a "simple decay"). We expect bouncing to happen when L2 regularization/ weight decay is important.
>
> 5/ We would also like to highlight that ABEL is better to track progress of the network performance during training (see fig S8j for example). We thought of doing similar experiments to what the author was proposing: run a bunch of experiments in a grid of hyperparameters spanning epochs, learning rate, weight decay and see which schedule is a better. However, we had a hard time about how to be honest with such experiments, given that we know the optimal parameters, if the grid doesn't include them we would be cheating but it the grid includes it then ABEL might not be as good.
>
> 6/ This is what section 4 is trying to convey: 1) L2 regularization is required for bouncing, 2) if either $\eta$ or $\lambda$ are too small the weight norm updates will have a fixed sign, 3) intuition why bouncing is beneficial: $\Delta |w|^2$ becomes smaller if there is no L2 and one decays the learning rate, if there is L2 (and one is near equilibrium after bouncing), decaying the learning rate yields a larger $\Delta |w|^2$. For scale invariant networks, larger $\Delta |w|^2$ implies more change in the angles -> model does not "get stuck", so maybe bouncing helps model explore more of the landscape, 4) a couple of experiments exploring properties of bouncing (described in section SM.C). We hope the reviewer can see some value on these observations.
>
> We would like to thank the reviewer again for they dedication to this paper and hope they would consider raising the score after reading this response.

---

### Official Review · Reviewer_iadJ · 2021-11-02

**Correctness:** 3
**Technical Novelty And Significance:** 2
**Empirical Novelty And Significance:** 2
**Recommendation:** 3
**Confidence:** 4

**Main Review:**

Pros:
* The proposed idea seems simple and effective. ABEL does not require lots of hyperparamters to tune and it is straightforward to implement.
* The empirical results looks adequate. To compare ABEL with other scheduling methods, the authors not only provide the final performance of model but also study the roustness of ABEL. ABEL gets close to baselines on several architectures and data sets, and it seems to be slightly more robust to the choices of initial learning rate and decay factor.
* Besides the ABEL algorithm itself, the authors also study bouncing weight norm in depth. They also show how to deal with the case where the weight norm is not bouncing and how it is connect to $l_2$ regularization. They also indicate that the presence of weight norm bouncing does not guarantee the schedule is beneficial but it depends on the complexity of the training task.

Cons:
* Some parts of the paper are not well-described or I could not find the related context. e.g. Algorithm 1 could have been more complete. Even the initialization of reached_minimum is not provided. The current one is readable but could be more detailed. Also, in section 4.1 equation (1) pops up from nowhere. As a reader, I find it hard to understand where this update of SGD comes from, so the authors might consider moving the deduction of that equation from supplements as it has only a few lines. Besides, the mathematics notations look weird to me and it would be helpful if the authors could give them more clear definitions.
* I haven't noticed any theoretical guarantee about such learning rate decay scheme. The results provided in the paper are most empirical. However, there are some parts not that obvious to me. For example, even if the authors say that learning rate schedules have advantages only when the training task is hard enough, I do not find any concrete evidence to support such view. Data augmentation might influence the results but it might be reason that the generalization ability of the model changes but not the training process gets affected. It would be helpful if the authors could give more insights about the reason for such observations.
* The experiments seem well designed, but ABEL is actually losing to Cosine for most cases. That somehow hurts the practicability of ABEL.

**Summary Of The Paper:**

The paper proposes a simple method to decay the learning rate automatically by tracking the weight norm of deep neural networks. A key observation from this work is that: there is a connection between weight norm bouncing, weight decay and optimizer convergence. The authors indicate that complex learning only benefits the training when weight norm bouncing exists, which further relies the presence of $l_2$ regularization on the weight. They also indicate that for those experiments without weight norm bouncing, like experiments in NLP and RL, the learning rate schedules could be simplified with no loss in performance.

Based on that, the paper proposes a new learning rate scheduler named ABEL. Basically, ABEL will drop the learning rate when the weight norm starts to bouncing. More specifically, when the weight norm growth slows down, the learning rate will be dropped by a pre-defined dropping factor.

**Summary Of The Review:**

The proposed idea is concise and interesting. However, it needs stronger justification to show that ABEL has advantage over other learning rate scheduling methods and the authors could provide a deeper study about the behavior of weight norm bouncing from a perspective of SGD dynamics.

---

> ### Author Response · Authors · 2021-11-09
> **Thanks for the review!**
>
> We are glad that you find the paper interesting and thank you for the detailed review. Find some comments below:
>
> . We have modified Algorithm 1 to be more detailed and address the reviewers concerns. We also made eq 4.1 self-contained and gave clearer definitions for the variables.
>
> . We indeed do not have theoretical guarantees, but to our knowledge there are no guarantees for convergence of SGD for real deep networks in general (other than large width or convex settings).
>
> . Our definition of "hard enough" (discussed at the end of section 3) is that tasks can't be easily memorized before the weight norm has bounced. Figure 4 shows an example of tasks where the training error hits one before weight norm bouncing. We spent a lot of time trying to come up with simple "toy" settings where complex learning schedules would be beneficial without success which is what prompted this point. We think it would be interesting to understand this further, and using the training accuracy as a proxy was the best probe we could think of.
>
> . Fair point about cosine, the main advantage of ABEL compared with cosine is that it can be run directly without knowing for how long the model should be evolved for. This is opposed to cosine decay, where for each time budget, one has to rerun the model. It is also more robust to hyperparameters. These two properties make it, in our opinion, a better default schedule for exploring novel settings. Related to the first point, one can also track training much better with ABEL than with cosine decay, for most models, cosine decay only beats ABEL late in training, making it really hard to know how good cosine will be, see figure S8j for a dramatic example.
>
> We hope to have addressed some of your concerns and would like to ask you to consider raising your score.

---

### Official Review · Reviewer_rqFL · 2021-11-08

**Correctness:** 3
**Technical Novelty And Significance:** 3
**Empirical Novelty And Significance:** 3
**Recommendation:** 6
**Confidence:** 3

**Main Review:**

Pros
1. The observation of weight norm bounce and its correlation with the effect of LR decay is novel (as far as I know) and interesting. The proposed technique is also simple enough to be practically useful.
2. I also appreciate the careful investigation on the generality of this finding, including experiments on different benchmarks as well as the importance of l2 weight decay. The results and analyses are pretty thought provoking.
3. Writing is pretty clear and easy to follow.

Cons & Questions
1. The title seems a bit too general (and bold), and can probably be made more specific about the findings.
2. As a paper about SGD, I'm a bit disappointed that training loss curves are not shown in the main text, along side test errors. Having both will help a lot on disentangling the benefit from optimization and generalization, which I think the paper should explicitly discuss and clarify.
3. Normalization layers (BN, LN) and others are also important confounding factors to the training dynamics, which I don't think the paper explicitly discuss. It appears to me that all models experimented are equipped with some normalization layers (correct me if I'm wrong), does removing them change the conclusion?
4. Other than the weight norm, I'm also curious how the gradient norm evolves during training and if it's useful in a similar way for guiding the LR decay.

***post rebuttal***
I'd like to thank the authors for addressing my questions. Overall, I decide to stand with my original ratings. I still believe that this paper presents interesting empirical findings regarding the empirical correlation between weight norm increase and the effectiveness of LR decay. And I think this observation can potentially help us better understand the training dynamics of deep models. However, I also share similar concerns with other reviewers, where the proposed method is ad hoc and does not strictly outperform standard LR decay schedules.

**Summary Of The Paper:**

This paper presents a novel method for automatically decaying the learning rate when training deep neural networks. The core idea relies on the observation that the weight norms often times bounce up during training, and that decaying the learning rate after the bouncing is beneficial. The authors conducted extensive experiments in different settings, highlighting the effectiveness of the proposed LR decay schedule, as well as the special cases where weight bouncing doesn't happen and simple decay schedules work competitively.  Overall, I think this paper shows intriguing findings, and opens potential new directions for further understanding the training and generalization of deep neural networks. However, there are also questions that I hope the authors can answer in the rebuttal period, in order to further improve this paper.

**Summary Of The Review:**

Overall I think this is an interesting work, and its quality can be further improved by answering the questions listed above.

---

> ### Author Response · Authors · 2021-11-10
> **Thanks for your time!**
>
> We thank you for the kind words and address your comments below. If you finds this response satisfactory, we would be extremely thankful if you were willing to raise your score.
>
> 2. We personally like losses too and were not sure if the public would appreciate them, since we did not find them super insightful. Added loss plots to SM.D.2. It seems to us like these different schedules have surprisingly the same generalization properties even if optimization-wise they are very different but we do not have anything smart to say about this.
>
> 3. We totally agree with normalization layers being possible confounders. Because of that we have VGG-16 experiments which exhibit the same weight bouncing phenomenon without BN (see table 1, figure S8), so normalization layers should not change our conclusion.
>
> 4. We did not find any interesting behavior in the gradient norm other than it being monotonically increasing and extremely noisy (as an example of this, see figure 2 of  https://arxiv.org/abs/2006.08419 for similar settings). Because the gradient norm is generically monotonically increasing independently of whether the weight norm bounces, we do not think one might draw similar conclusions from it (bouncing seems important for schedules to be beneficial).
>
> Thanks again for the time spent reviewing this submission.

---

### Decision · Program_Chairs · 2022-01-20

**Decision:**

Reject

**Comment:**

The authors provide an investigation into tuning learning rate schedules. The problem is certainly of great practical importance. After discussion, the reviewers felt the main idea of the paper is worth pursuing, but could use significant refinement. One reviewer suggests: "
"...better treatment of the background material, clearer identification on when the weight norm behaviour happens beside norm (possibly looking also for counter-examples!), rethinking section 6, and a more convincing set of experiments (for showing convincing evidence about e.g. 5.2). Regarding this last point, I want to clarify that in my review I mentioned [1] not for the grid search, but rather for the time-controlled experiments. If you go with random search for selecting the hyperparameters of the learning rate adaptation methods. I personally think that a recipe to make the comparison fair enough is to choose a prior distribution (e.g. uniform/log uniform) that covers reasonable values (e.g. as used for different datasets) with mean equal/close to the known well-performing ("optimal") value." Other reviewers were generally of a similar opinion. The authors are encouraged to continue with the work, taking reviewer comments into account for updated versions.